# Late-Onset Hypogonadism in a Male Patient with Long COVID Diagnosed by Exclusion of ME/CFS

**DOI:** 10.3390/medicina58040536

**Published:** 2022-04-13

**Authors:** Yoshiaki Soejima, Yuki Otsuka, Kazuki Tokumasu, Yasuhiro Nakano, Ko Harada, Kenta Nakamoto, Naruhiko Sunada, Yasue Sakurada, Kou Hasegawa, Hideharu Hagiya, Keigo Ueda, Fumio Otsuka

**Affiliations:** Department of General Medicine, Okayama University Graduate School of Medicine, Dentistry and Pharmaceutical Sciences, Okayama 700-8558, Japan; soesoejima0121@gmail.com (Y.S.); otsuka@s.okayama-u.ac.jp (Y.O.); tokumasu@okayama-u.ac.jp (K.T.); y-nakano@okayama-u.ac.jp (Y.N.); me422084@s.okayama-u.ac.jp (K.H.); jn4okx@gmail.com (K.N.); naru.kun.red.1117@gmail.com (N.S.); sakurada202@gmail.com (Y.S.); khasegawa387@yahoo.co.jp (K.H.); hagiya@okayama-u.ac.jp (H.H.); kuedampo@okayama-u.ac.jp (K.U.)

**Keywords:** general fatigue, free testosterone, late-onset hypogonadism, long COVID, post-COVID condition

## Abstract

After the acute phase of COVID-19, some patients have been reported to have persistent symptoms including general fatigue. We have established a COVID-19 aftercare clinic (CAC) to provide care for an increasing number of these patients. Here, we report the case of a 36-year-old man who developed post-COVID fatigue after acute infection with SARS-CoV-2. In the acute phase of COVID-19, the patient’s fever resolved within four days; however, general fatigue persisted for three months, and he visited our CAC 99 days after the initial infection. Examination revealed a high Aging Male’s Symptoms (AMS) score of 44 and low free testosterone (FT) level of 5.5 pg/mL, which meet the Japanese criteria of late-onset hypogonadism (LOH) syndrome. Imaging studies revealed an atrophic pituitary in addition to fatty liver and low bone mineral density. Anterior pituitary function tests showed a low follicle-stimulating hormonelevel and delayed reaction of luteinizing hormone (LH) after gonadotropin-releasing hormone (GnRH) stimulation, indicating the possibility of hypothalamic hypogonadism in addition to primary hypogonadism seen in patients with post-COVID-19 conditions. After the initiation of Japanese traditional medicine (Kampo medicine: hochuekkito followed by juzentaihoto), the patient’s symptoms as well as his AMS score and serum FT level were noticeably improved. Furthermore, follow-up tests of GnRH stimulation revealed improvements in LH responsiveness. Although many patients have been reported to meet the criteria of ME/CFS such as our case, we emphasize the possibility of other underlying pathologies including LOH syndrome. In conclusion, LOH syndrome should be considered a cause of general fatigue in patients with post-COVID-19 conditions and herbal treatment might be effective for long COVID symptoms due to LOH (264 words).

## 1. Introduction

Coronavirus disease 2019 (COVID-19) has been reported to cause persistent symptoms even after the acute phase of infection [1,2]. The commonly reported symptoms include general fatigue, shortness of breath, dysgeusia, dysosmia, insomnia and alopecia [1,3,4,5]. In 2021, the World Health Organization defined these sequalae as a “post-COVID-19 condition” [6]. It has been reported that at least one-third of patients with COVID-19 conditions suffer from these persistent symptoms in the chronic phase [1,2,3,5].

Due to the increasing number of patients with persistent symptoms, we established a COVID-19 aftercare clinic (CAC) at Okayama University Hospital in February 2021 and we have been providing comprehensive care and treatment. Our previous study revealed that the most frequent general symptoms in these patients were general fatigue, dysosmia, dysgeusia, alopecia, headache, dyspnea and dyssomnia, with general fatigue being the most frequent symptom [7]. We also reported that the numbers of patients with the remaining symptoms had decreased to about half at 3 months after the initial visit [8], while about half of the patients still suffered from persistent symptoms. Several disorders of endocrine organs including the hypothalamus, pituitary gland, adrenal gland, thyroid gland and genital gland have been suggested to be involved in the mechanism of the post-COVID-19 condition [9]. However, information on the phenomena occurring at the molecular and cellular levels is still limited, and therapeutic strategies have not yet been established [10].

Here, we report on the long-term follow-up and recovery of a case of late-onset hypogonadism (LOH) syndrome in a patient who suffered from general fatigue after COVID-19 infection. This case provides important information for characterizing the fatigue symptoms seen in post-COVID condition and may provide a breakthrough for a possible strategy for treatment of post-COVID fatigue.

## 2. Case Presentation of a Male Patient with Post-COVID Conditions

A 36-year-old man who had a family member diagnosed with COVID-19 developed a fever of 38.5 °C, and a polymerase chain reaction test showed a positive result for severe acute respiratory syndrome coronavirus 2 (SARS-CoV-2). His fever was completely resolved within four days; however, general fatigue gradually developed and persisted even after the isolation period. He was referred from an online clinic to our CAC for further evaluation 99 days after the initial symptoms of COVID-19. He did not complain of any symptoms other than insomnia including headache, dysgeusia, dysosmia and alopecia. He had no particular past medical history or familial history, but he was taking escitalopram oxalate, brotizolam and clotiazepam for the insomnia that developed after COVID-19 infection. He was a smoker with 7.5 pack years but not an alcohol drinker.

Upon admission, he weighed 63.9 kg, with a normal body mass index of 21.5 (kg/m^2^). His vital signs were as follows: heart rate, 83 beats per minute; blood pressure, 135/72 mmHg; body temperature, 36.3 °C; and respiratory rate, 16 per minute with an O_2_ saturation of 95% under ambient air. Upon physical examination, he had no specific signs indicating endocrine disorders such as thyroid goiter, skin pigmentation, hair loss, edema or other abnormal findings. His self-rating depression scale (SDS) score was 47, which is categorized as “mildly depressed” [11]. His frequency scale for the symptoms of gastroesophageal reflux disease (FSSG) was 35. His Aging Male’s Symptoms (AMS) scale was 44 out of 85, indicating moderate LOH syndrome [12]. Routine laboratory examinations including blood cell counts and biochemistries were found to be all within normal ranges; however, of note, hormonal evaluation revealed that serum-free testosterone (FT) level was decreased to 5.5 pg/mL, which is sufficiently low to meet the Japanese criteria of late-onset hypogonadism (LOH) syndrome [13] (Table 1).

Based on the diagnosis of LOH syndrome, we performed additional examinations for the primary etiology and complications of LOH syndrome. Abdominal ultrasonography and computed tomography (CT) revealed fatty liver but no abnormalities in the bilateral adrenal glands. Magnetic resonance imaging (MRI) revealed partially empty sella in the anterior pituitary region, while the pituitary stalk and T1-hyperintensity of the posterior pituitary gland were normal (Figure 1). Anterior pituitary function tests were then performed. The tests showed that serum follicle-stimulating hormone (FSH) levels were significantly low (up to 8.3 mIU/mL) and luteinizing hormone (LH) secretion showed a delayed reaction in response to gonadotropin-releasing hormone (GnRH) injection (Figure 2A). On the other hand, the anterior pituitary responses to corticotropin-releasing hormone (CRH) and thyrotropin-releasing hormone (TRH) were almost normal (Figure 2B,C). In addition, bone mineral density (BMD) in the lumbar spine and that in the femur measured by dual energy X-ray absorptiometry were 73% and 87%, respectively, of the young adult means.

The findings suggested hypothalamic hypogonadism as an etiology of his long-lasting fatigue. Increased visceral fat and slightly decreased BMD were considered possible complications of LOH syndrome. After the initiation of treatment with hochuekkito, a Japanese traditional medicine (Kampo medicine), his symptom and AMS score gradually improved in accordance with the increase in serum FT. After the administration of hochuekkito for about 2 months, the administration of another Kampo medicine, juzentaihoto, was started as we found the medical signs of ‘qi deficiency’ and ‘blood deficiency’, in which vital energy and the circulation system are in poor condition, including coldness of the extremities (Figure 3A). Repeated GnRH tests at about one month (phase 2) and six months (phase 3) after the first examination (phase 1) showed that LH responsiveness to GnRH was gradually improved (Figure 3B), whereas the FSH peak response to GnRH was not changed during the clinical course (Figure 3C).

## 3. Discussion

Here, we reported a unique case of LOH syndrome that was diagnosed and followed up by careful investigation for persistent general fatigue after COVID-19. An LOH condition often affects multiple organs and may cause erectile dysfunction, depression, obesity, osteoporosis, anemia, and impairment of insulin resistance as well as general fatigue [13,14]. This case indicates the possibility of latent hypogonadism being related to general fatigue in patients with the post-COVID-19 condition.

One possible pathogenesis of post-COVID-19 LOH is primary hypogonadism [15,16,17]. SARS-CoV-2 is known to target and enter host cells via angiotensin-converting enzyme 2 (ACE2) receptors [18], and Leydig cells in the testis are known to express ACE2 receptors abundantly [19]. In addition, SARS-CoV-2 was detected from the testes of cadavers of people with COVID-19 infections [20]. Therefore, the immune response and the following oxidative stress of the testes to SARS-CoV-2 are considered a trigger for primary hypogonadism [15,16,17,19,21].

Moreover, considering the LH responsiveness to GnRH administration in this case, hypothalamic hypogonadism is also likely to be involved in the pathology of hypogonadism. It has been shown that direct infiltration of SARS-CoV-2 to the hypothalamus or the pituitary activates an autoimmune response and then induces an ME/CSF-like condition [9]. It has not yet been elucidated what occurred in the hypothalamus and pituitary in patients with the long COVID condition. The secondary empty sella is, in general, known to develop after treatment of pituitary tumors, after spontaneous necrosis of pituitary adenomas, after brain trauma, and after pituitary infectious processes or autoimmune diseases [22]. The MRI in our case newly suggested the existence of empty sella or atrophic pituitary in patients with the post-COVID condition.

On the other hand, general fatigue, which is one of the characteristic symptoms of LOH syndrome, is also a common symptom of myalgic encephalomyelitis/chronic fatigue syndrome (ME/CFS) [23,24,25]. There are several reports showing similarities between long COVID and ME/CFS [26], and up to 13 to 45 percent of patients with long COVID met the ME/CFS criteria [26,27,28,29]. Indeed, the present patient suffered from post-exertional malaise, sleep disturbance and ‘lack of stamina’ compared with his pre-illness condition, all of which are included in the criteria of ME/CFS [23,24,25]. However, we stress here the possibility of other latent pathologies, including LOH syndrome, in patients who have been conventionally and/or temporarily diagnosed as having ME/CFS after COVID-19 infection.

Androgen replacement therapy (ART) and treatment with phosphodiesterase-5 inhibitors, antidepressants or herbal medicines (Kampo) are possible treatment options for LOH syndrome [13,30]. Since our patient was under 40 years of age, an age when ART is not usually applicable, herbal treatment was attempted. It has been reported that herbal medicine improved AMS physiological factors compared with the baseline [30]. Another study showed that 8-week administration of hochuekkito, which is a Kampo medicine prescribed in our case, significantly increased serum FT level [31]. Juzentaihoto is another Kampo medicine used for treatment of fatigue in patients with systemic wasting diseases [32], and treatment with juzentaihoto was initiated in our patient because of signs of insufficient amounts of blood (‘blood deficiency’) in addition to insufficient quantities of qi (‘qi deficiency’). It is of note that a wide variety of symptoms including fatigue, depressive mood, metabolic disorders and even gastroesophageal reflux symptoms might indicate the usefulness of measuring serum FT level in patients with post-COVID-19 conditions [33] and that Kampo medicines might be effective in patients with post-COVID-19 conditions [34].

Although post-COVID-19 symptoms have been well studied, most of the researchers have targeted and followed up with patients with severe illness treated in an inpatient setting. However, since the majority of patients with COVID-19 infections have mild diseases [35], it is necessary to focus on persistent symptoms after mild to moderate COVID-19 that are treated in outpatient settings. Symptoms persisting after mild COVID-19 infections result in impairment of work and daily functioning [36]. Therefore, we emphasize the importance of careful investigation and appropriate examinations for prolonged general fatigue even after mild COVID-19 infections in order to establish therapeutic strategies and to improve the quality of life for patients with long COVID.

In conclusion, we reported a proven case of LOH syndrome after mild COVID-19 infection. LOH syndrome should be distinguished from ME/CFS in patients complaining of persistent malaise, which is the most frequent symptom in the post-COVID-19 condition. Kampo medicine might be a treatment option for patients with LOH even after COVID-19 infection.

## Figures and Tables

**Figure 1 medicina-58-00536-f001:**
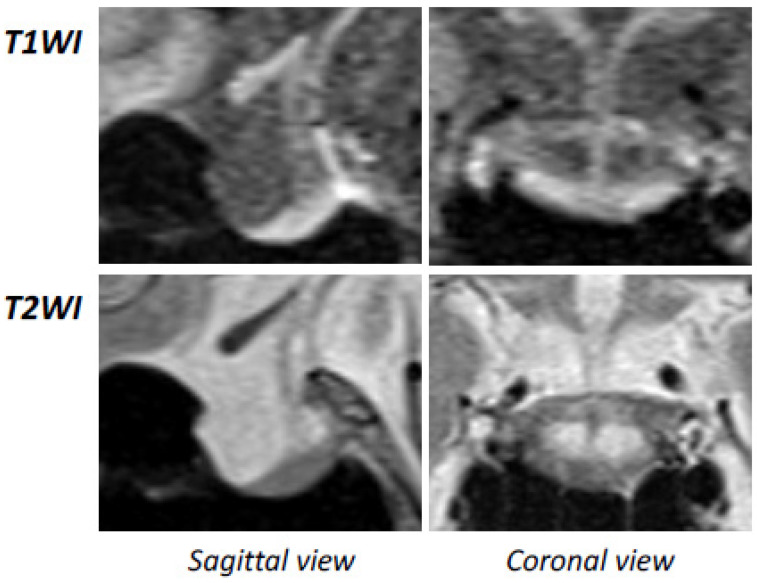
Pituitary magnetic resonance imaging (MRI). An atrophic pituitary indicating partially empty sella was demonstrated by brain MRI, while the pituitary stalk and posterior gland were intact. T1WI and T2WI: T1- and T2-weighted images.

**Figure 2 medicina-58-00536-f002:**
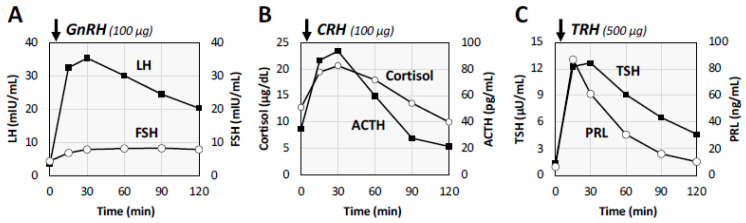
Anterior pituitary function tests. Serum levels of follicle-stimulating hormone (FSH) were consistently low, and the secretory reactions of luteinizing hormone (LH) were delayed after gonadotropin-releasing hormone (GnRH) stimulation (**A**). Corticotropin-releasing hormone (CRH) test (**B**) and thyrotropin-releasing hormone (TRH) test (**C**) showed normal responses.

**Figure 3 medicina-58-00536-f003:**
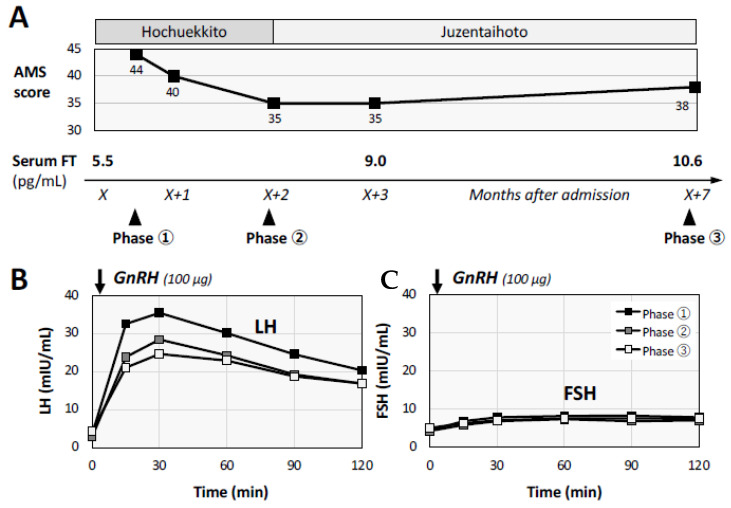
Clinical course and alteration of gonadotropin responses. The patient’s clinical course after administration of herbal medicines is shown with Aging Male’s Symptoms (AMS) scores, serum free testosterone (FT) levels and the results of gonadotropin-releasing hormone (GnRH) tests (**A**). The symptoms, AMS score and serum FT level were gradually improved simultaneously. Based on the results of repeated GnRH tests at three different phases, luteinizing hormone (LH) responsiveness to GnRH was improved (**B**), while the peak responses of follicle-stimulating hormone (FSH) were not apparently changed during the observation period (**C**).

**Table 1 medicina-58-00536-t001:** Laboratory data upon admission.

Complete Blood Count				Biochemistry		
White blood cells	7610		/μL	Total protein	7.6	g/dL
Neutrophils	73.7		%	Albumin	5.0	g/dL
Lymphocytes	20.8		%	Total bilirubin	1.17	mg/dL
Eosinophils	0.6		%	Aspartate transaminase	16	U/L
Red blood cells	550 × 10^4^		/μL	Alanine transaminase	16	U/L
Hemoglobin	15.7		g/dL	Alkaline phosphatase	68	U/L
Platelets	28 × 10^4^		/μL	γ-glutamyl transpeptidase	21	U/L
Endocrine data [Normal Range]				Lactate dehydrogenase	178	U/L
Cortisol	12.9	[4.5–21.1]	μg/dL	Sodium	141	mmol/L
Adrenocorticotropin	44.1	[7.2–63.3]	pg/mL	Potassium	3.6	mmol/L
Free thyroxine	1.48	[0.97–1.69]	ng/dL	Chloride	106	mmol/L
Thyroid-stimulating hormone	1.40	[0.33–4.05]	μIU/mL	Calcium	9.7	mg/dL
Follicle-stimulating hormone	4.2	[1.3–17.0]	μIU/mL	Phosphate	3.2	mg/dL
Lutenizing hormone	3.0	[0.52–7.8]	μIU/mL	Magnesium	2.1	mg/dL
Prolactin	13.7	[3.0–17.3]	ng/mL	Zinc	89	μg/dL
Growth hormone	2.54	[0–2.47]	ng/mL	Blood urea nitrogen	8.0	mg/dL
Insulin-like growth factor 1	258	[99–275]	ng/mL	Creatinine	0.80	mg/dL
**Free testosterone**	5.5	[6.5–17.7]	pg/mL	Uric acid	5.6	mg/dL
vitamin B1	51	[24–66]	ng/mL	Triglyceride	102	mg/dL
vitamin B12	547	[197–771]	pg/mL	Low-density lipoprotein cholesterol	123	mg/dL
Fasting plasma glucose	104	[73–109]	mg/dL	C-reactive protein	0.02	mg/dL

## Data Availability

No new data were created or analyzed in this study. Data sharing is not applicable to this article.

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
