# Peer review of "Late-Onset Hypogonadism in a Male Patient with Long COVID Diagnosed by Exclusion of ME/CFS"

_medicina, 2022, doi:10.3390/medicina58040536_

Round 1

Reviewer 1 Report

What authors claim ME/CFS should be identified or clarified.

I would suggest to cite 2011 ME International Consensus Criteria paper (Carruthers et al ).

Author Response

RE: Manuscript ID: medicina-1663964-R1

Medicina: Special Issue "Advances in ME/CFS Research and Clinical Care"

Title: Possibility of late-onset hypogonadism in a male patient with long COVID mimicking ME/CFS: Soejima Y., et al.

Dear Editor of Medicina:

Dr. Daisy Daisy Tang: Section Managing Editor

 We would like to thank the editor for the careful review and favorable comments on the manuscript.  According to the editor’s and referees’ comments and questions, we revised our manuscript as addressed in the following sections.  We would also like to thank you for allowing us to resubmit our revised manuscript.  Also, the revised manuscript underwent English proofreading again.  We hope that the revised manuscript is now acceptable for publication in Medicina.

Sincerely yours,

Yoshiaki Soejima, M.D.

Fumio Otsuka, M.D., Ph.D.

Reviewer 2 Report

The general study design and other details pertaining to long COVID or LOH syndrome are fine - this is a straightforward clinical examination of one individual. But I have serious objections regarding the suggested connection to ME/CFS and the author's introductions and conclusions pertaining to ME/CFS.

Particularly:

"Abstract: After the acute phase of coronavirus disease 2019 (COVID-19) infection, some patients have been reported to have persistent symptoms including general fatigue mimicking myalgic encephalomyelitis/chronic fatigue syndrome (ME/CFS). "

This is misleading, as ME/CFS is not characterised by general fatigue. ME/CFS is characterised by two more highly specific fatigue-related symptoms (required for diagnosis under up-to-date case definitions) and a third suite of variable symptoms (ie: not merely "general fatigue"):

1) Post exertional malaise (worsening of symptoms following exertion). Can be delayed and often termed "payback".

2) Fatigue less aided by rest, and that is disproportionately high compared to the fatigue-inducing action.

3) a constellation of varied symptoms across multiple or all body systems which is likely to vary with the individual + mode of onset. 

The title of the paper is similarly misleading. It does not mimic ME/CFS. The title must be changed in light of this.

Up-to-date case definitions which require post-exertional malaise should be referred to in accordance with the increasingly accepted international view. eg: International Consensus Criteria (ICC, Myalgic encephalomyelitis: International Consensus Criteria - Carruthers - 2011 - Journal of Internal Medicine - Wiley Online Library), Canadian Consensus Criteria (CCC, Myalgic Encephalomyelitis/Chronic Fatigue Syndrome: Clinical Working Case Definition, Diagnostic and Treatment Protocols: Journal of Chronic Fatigue Syndrome: Vol 11, No 1 (tandfonline.com)). It must be emphasized that ME/CFS is a specific combination of symptoms and onset pattern that is distinct from general fatigue. Theorised links between long-covid and ME/CFS should be outlined in detail in the text. At both the symptomatic and biological levels, specific examples of similarities must be described with specific details in the text itself which relate to the current study. A simple referencing of the review paper ref [9] without detail is insufficient to establish this connection. 

The authors write: "In conclusion, we reported a proven case of LOH syndrome after mild COVID-19. LOH syndrome may underlie and mimic ME/CFS in patients complaining of general fatigue, which is the most frequent symptom in the post COVID-19 condition."

What is the basis for this claim? One long-covid patient with LOH syndrome is not sufficient to claim that LOH syndrome may underlie and mimic ME/CFS. For one, the link between long-covid and ME/CFS is not yet demonstrated clearly. It is theorised and expected in at least a subset of cases, yes, but yet unproven as relevant studies in the ME/CFS are currently still underway. More importantly, if this was a case of LOH syndrome in specifically an ME/CFS patient and not a long-covid patient, how would one occurrence of a particular comorbidity reveal any mechanism underlying the disease? This claim must be removed from the paper in the absence of direct causal investigation between ME/CFS molecular disease mechanisms and LOH syndrome. In this case, where the patient does not have confirmed ME/CFS and no direct causal investigation has been undertaken at the molecular level, it is entirely inappropriate to make this claim. The claim, as it stands, is wholly unjustified and muddies the waters of the ME/CFS field which is in need of clarity and highly-detailed causal molecular studies.

Overall, the links between this study and ME/CFS are restricted to the (yet unproven) links between long-covid and ME/CFS. The nature of this link - both including the specific similarities and this particular limitation must be addressed clearly in the text. Otherwise, this paper which is an interesting case-study of long-covid and LOH syndrome does not bear much relevance to ME/CFS as the authors claim, thus neither should it fall within the remit of an ME/CFS special issue without major amendment to demonstrate the relevance.

Author Response

(The authors gave the same response as above.)

Round 2

Reviewer 2 Report

Thank you to the authors for kindly including the suggested changes. This is now a very interesting and informative paper. Congratulations to the authors.